# The Effect of Face Mask Use on COVID-19 Models

**Fan Bai** [1,*,†] **and Fred Brauer** [2,†]

1   Hausdorff Center for Mathematics, University of Bonn, 53115 Bonn, Germany
2   Department of Mathematics, University of British Columbia, Vancouver, BC V6T 1Z2, Canada;
    brauer@math.ubc.ca or brauer@math.wisc.edu
*   Correspondence: bai@hcm.uni-bonn.de
†   Both authors contributed equally to this work.

**Abstract:** We begin with a simple model for the COVID-19 epidemic and add face mask usages and testing and quarantine of infectives. We estimate the effect on the reproduction number and discuss the question of whether the epidemic can be controlled by increased use of face masks.

**Keywords:** COVID-19; deterministic modeling; face mask usages; test and quarantine; reproduction number; final epidemic size





## 1. Introduction

The outbreak of a new coronavirus in Wuhan, China, now named COVID-19, has led to many efforts to formulate models to describe its spread and estimate its outcome. The government of the US was disturbed greatly by news of a model from Imperial College, London [1] predicting that in the absence of any control measures, there could be as many as 22,000,000 coronavirus deaths in the US. A substantial portion of the US response to the coronavirus response has been a switch to a quite different model [2], predicting considerably fewer deaths. At this writing there have been 262,000 deaths in the US, and there is an outbreak that is out of control and is adding more than 1000 deaths per day.

The coronavirus COVID-19 is particularly difficult to model because there appear to be many asymptomatic cases and individuals may become infectious before the appearance of symptoms. A detailed compartmental model would require susceptible, exposed, presymptomatic, infectious, and asymptomatic individuals. To identify infectives in COVID-19, it is necessary to carry out a test because of the significant number of asymptomatic cases and cases with very mild symptoms. This suggests using testing to identify infectives and ignoring the presence or absence of symptoms in assigning individuals to a compartment. Thus, we can use an *SEIR* model, identifying presymptomatic exposed individuals as infective,

$$
\begin{aligned}
S' &= -\beta \frac{S}{N} I \\
E' &= \beta \frac{S}{N} I - \eta E \\
I' &= \eta E - \gamma I \\
R' &= \gamma I,
\end{aligned}
\tag{1}
$$

or we can separate asymptomatic infectives in an *SEIAR* model

$$S' = -\beta \frac{S}{N} I - \delta \frac{S}{N} A$$
$$E' = \beta \frac{S}{N} I + \delta \frac{S}{N} A - \eta E$$
$$I' = f\eta E - \gamma I \tag{2}$$
$$A' = (1 - f)\eta E - \theta A$$
$$R' = \gamma I + \theta A.$$

In model (2), a fraction $f$ of exposed members become symptomatic, and infective and asymptomatic individuals may have different infectivities and recovery rates. However, the available data suggest that infectivity and recovery rates of asymptomatic individuals are approximately the same as for symptomatic individuals. This implies that we can make estimates without distinguishing between symptomatic and asymptomatic individuals. This is fortunate, since estimates of the fraction of disease cases that are asymptomatic range from 5% to 85%.

The basic reproduction number for the model (1) is $\mathcal{R}_0 = \beta/\gamma$. It is believed that the basic reproduction number for COVID-19 in the US is approximately 2.5, and the mean infective period is approximately 5 days. This leads to the estimates $\beta = 0.5$, $\gamma = 0.2$. Estimates of the exposed period suggest a mean of 14 days, which would lead to an estimate $\eta = 0.07$.

We applied model (1) using population sizes $I(0) = 100$, $N = 325,000,000$ and parameter values $\beta = 0.5$, $\gamma = 0.2$, so that $\mathcal{R}_0 = 2.5$ to estimate the possible size of an epidemic in the United States. We obtain a number 290,000,000 of disease cases for the United States if no control measures are used. This compares to a number 263,000,000 of cases obtained from the much more detailed model [1] of the Imperial College London, which was viewed as a worst-case scenario by the US government. We suggest that this helps to validate the use of model (1) as an approximation to reality.

In fact, the suggestion that if no control measures are used, there could be as many as 263,000,000 disease cases resulting in 2,000,000 deaths is highly unlikely. During the Ebola outbreak in western Africa in 2014-2016, the spread of disease was much slower than would have been expected from estimates of the basic reproduction number. There is considerable anecdotal evidence that many people altered their behavior, especially with respect to funeral practices for disease victims, and that this produced significantly fewer disease cases than expected, even before the start of any governmental efforts to control disease. Even earlier, during the "Spanish flu" pandemic of 1918-1919, some cities banned large gatherings and experienced much lower case counts. Various models predicting slower growth have been proposed to give a better match to data [3–5]. While it has been suggested that early response to the pandemic by the US government saved more than 2,000,000 lives, this is clearly unfounded.

It is argued by some that a great deal of harm is being done to the economy by the social distancing measures that have been recommended, and that it important to assist the recovery of the economy by loosening the social distancing restrictions. Many states began loosening distancing recommendations while the number of COVID-19 cases was still increasing, contrary to the recommendations of public health professionals. As the end of the year 2020 approaches, there are many signs of pandemic fatigue. Organized opposition to social distancing regulations is apparent, and recommendations of face mask use are being interpreted as political statements. The holiday season has led to substantial increases in travel to family gatherings against the recommendations of public health professionals, followed by substantial increases in COVID-19 infections.

We are being informed that the observed increase in COVID-19 cases is caused by increased testing rates, and it is even being argued that the number of tests should be decreased to assist in control. It might be pointed out that decreased testing does not imply

a decrease in the number of cases; cases that are not found are still cases that would lead to more cases in the future.

## 2. The Effect of Face Mask Use

Classically, the only treatment methods available for outbreaks of new diseases are isolation of diagnosed infectives and quarantine of suspected infectives who may be identified by contact tracing. A newer treatment approach is the reduction of contacts. Currently, many places have recommended restrictions on public gatherings and closing of many businesses and have advised people who are particularly at risk because of age or pre-existing health conditions to stay at home. Many districts have closed schools and day-care centres to try to protect children. Many companies have encouraged some of their employees to work from home. All of these measures have the effect of reducing contacts and preventing new infections, and this is what is meant by "social distancing". We will study the effects of social distancing elsewhere; here, we build social distancing into our model only by assuming that a fraction of the population wears face masks when in close proximity to others, and that the use of face masks decreases both infectivity and susceptibility to infection. The effect of including the use of face masks in the model has also been studied in [6]; we cite some results on final size relations obtained there. For complete studies on final epidemic size relations in either homogeneous or heterogeneous mixing populations, we refer to Sections 2.4 and 5.3 in [7].

To study the impact of face mask use, we consider that the population is divided into two groups, based on whether individuals are complying with the rule of wearing masks in public. Group 1 members, who are mask wearers, form a fraction $\rho_2$ of the total population. Members of both groups have a contact rate $a$. Mixing between groups is assumed proportionate ([8–10]). The fractions of contacts of susceptibles with groups 1 and 2 are

$$p_1 = \rho_2, \quad p_2 = 1 - \rho_2.$$

It is noted that, for each contact between individuals of whom one has been infected and the other is susceptible, there is a probability that infection will actually be transmitted. This probability depends on many factors, such as the infectivity of the member who has been infected and the susceptibility of the susceptible member. We assume that wearing masks reduces infectivity by $\sigma_2$ and susceptibility by $\sigma_3$. Thus, the epidemic model is

$$
\begin{aligned}
S_1' &= -\sigma_3 \beta S_1 \left[ \sigma_2 p_1 \frac{I_1}{N_1} + p_2 \frac{I_2}{N_2} \right], \\
S_2' &= -\beta S_2 \left[ \sigma_2 p_1 \frac{I_1}{N_1} + p_2 \frac{I_2}{N_2} \right], \\
E_1' &= \sigma_3 \beta S_1 \left[ \sigma_2 p_1 \frac{I_1}{N_1} + p_2 \frac{I_2}{N_2} \right] - \eta E_1, \\
E_2' &= \beta S_2 \left[ \sigma_2 p_1 \frac{I_1}{N_1} + p_2 \frac{I_2}{N_2} \right] - \eta E_2, \\
I_1' &= \eta E_1 - \gamma I_1, \\
I_2' &= \eta E_2 - \gamma I_2, \\
R' &= \gamma (I_1 + I_2).
\end{aligned}
\tag{3}
$$

To calculate the reproduction number using the next-generation matrix method [11], we write

$$
F = \begin{bmatrix} \sigma_3 \sigma_2 \beta p_1 & \sigma_3 \beta p_2 \frac{N_1}{N_2} \\ \sigma_2 \beta p_1 \frac{N_2}{N_1} & \beta p_2 \end{bmatrix} = \begin{bmatrix} \sigma_3 \sigma_2 \beta p_1 & \sigma_3 \beta p_1 \\ \sigma_2 \beta p_2 & \beta p_2 \end{bmatrix},
\tag{4}
$$

and

$$
V = \begin{bmatrix} \gamma & 0 \\ 0 & \gamma \end{bmatrix}.
\tag{5}
$$

Then, the next-generation matrix is

$$FV^{-1} = \begin{bmatrix} \frac{\sigma_3\sigma_2\beta p_1}{\gamma} & \frac{\sigma_3\beta p_1}{\gamma} \\ \frac{\sigma_2\beta p_2}{\gamma} & \frac{\beta p_2}{\gamma} \end{bmatrix}.$$

This matrix has determinant zero. Thus, the non-zero eigenvalue is the trace of the matrix, and

$$\mathcal{R}_0 = [\sigma_3\sigma_2\rho_2 + (1-\rho_2)]\frac{\beta}{\gamma} = [1 + (\sigma_2\sigma_3 - 1)\rho_2]\frac{\beta}{\gamma}. \tag{6}$$

Thus, the resulting reduction factor $\Gamma_2$ in the reproduction number in (6) is given by

$$\Gamma_2 = 1 + (\sigma_2\sigma_3 - 1)\rho_2. \tag{7}$$

Since $\sigma_2 \leq 1$, $\sigma_3 \leq 1$, $\Gamma_2$ is monotone decreasing when $\rho_2$ increases. There is evidence that wearing masks can dramatically lower the susceptibility and infectivity. The reduction factor $\Gamma_2$ depends strongly on the type of mask used; we assume here that $\sigma_3 = 0.75$, $\sigma_2 = 0.25$. Then, the reduction factor $\Gamma_2$ is about 0.63 if the fraction of mask wearers is the currently assumed value of 50%. If the fraction of mask wearers could be increased to 74%, the corresponding reduction factor would decrease to 0.39, and this would suffice to decrease the assumed value 2.5 of the basic reproduction number of the model (1) below 1, thus controlling the epidemic.

As shown in [6], the final epidemic size can be expressed in the following equations,

$$\begin{aligned} \log \frac{S_1(0)}{S_1(\infty)} &= \frac{\sigma_2\sigma_3\beta\rho_2}{\gamma}\left[1 - \frac{S_1(\infty)}{N_1}\right] + \frac{\sigma_3\beta(1-\rho_2)}{\gamma}\left[1 - \frac{S_2(\infty)}{N_2}\right], \\ \log \frac{S_2(0)}{S_2(\infty)} &= \frac{\sigma_2\beta\rho_2}{\gamma}\left[1 - \frac{S_1(\infty)}{N_1}\right] + \frac{\beta(1-\rho_2)}{\gamma}\left[1 - \frac{S_2(\infty)}{N_2}\right]. \end{aligned} \tag{8}$$

The final size relation further indicates that

$$\frac{S_1(\infty)}{S_1(0)} = \left(\frac{S_2(\infty)}{S_2(0)}\right)^{\sigma_3}. \tag{9}$$

Because $\sigma_3 < 1$, we have

$$\frac{S_1(\infty)}{S_1(0)} > \frac{S_2(\infty)}{S_2(0)}. \tag{10}$$

This inequality leads to the conclusion that the attack rate for group 1 is smaller than that for group 2 due the lower level of susceptibility in group 1. The attack rate for a group is defined as the ratio of the total number of infected individuals and the population size of the group. Thus, wearing face masks has a very positive impact on containing the spread of the epidemic and protecting people from being infected.

We performed numerical experiments to simulate the model (3), show the final size relation, and compare the attack rates for two groups. The parameter values for simulations are presented in Table 1.

**Table 1.** Parameter values.

| Parameter | Numeric Value |
|---|---|
| $\sigma_3$ | 0.75 |
| $\sigma_2$ | 0.25 |
| $\beta$ | 0.5 |
| $\eta$ | 0.07 |
| $\gamma$ | 0.2 |
| $p_1$ | $\rho_2$ |
| $p_2$ | $1 - \rho_2$ |

The reduction factor $\Gamma_2 = 1 - \frac{13}{16}\rho_2$ can be calculated for the assumed value of $\rho_2$. If $\rho_2 > 0.74$, the reproduction number can be decreased below 1. We consider two scenarios $\rho_2 = 0.2$ and $\rho_2 = 0.5$, which represent the notion that only a small proportion of the population is willing to wear masks, and around half of the population is following the rule. The population size is assumed to be $N = 5000$ for the purpose of simplicity, and initially there are only a few infection cases. The simulations are presented in Figure 1. It is observed that the higher the proportion of face mask use, the smaller the value of the reproduction number, which has a significant impact on flattening the epidemic curve.

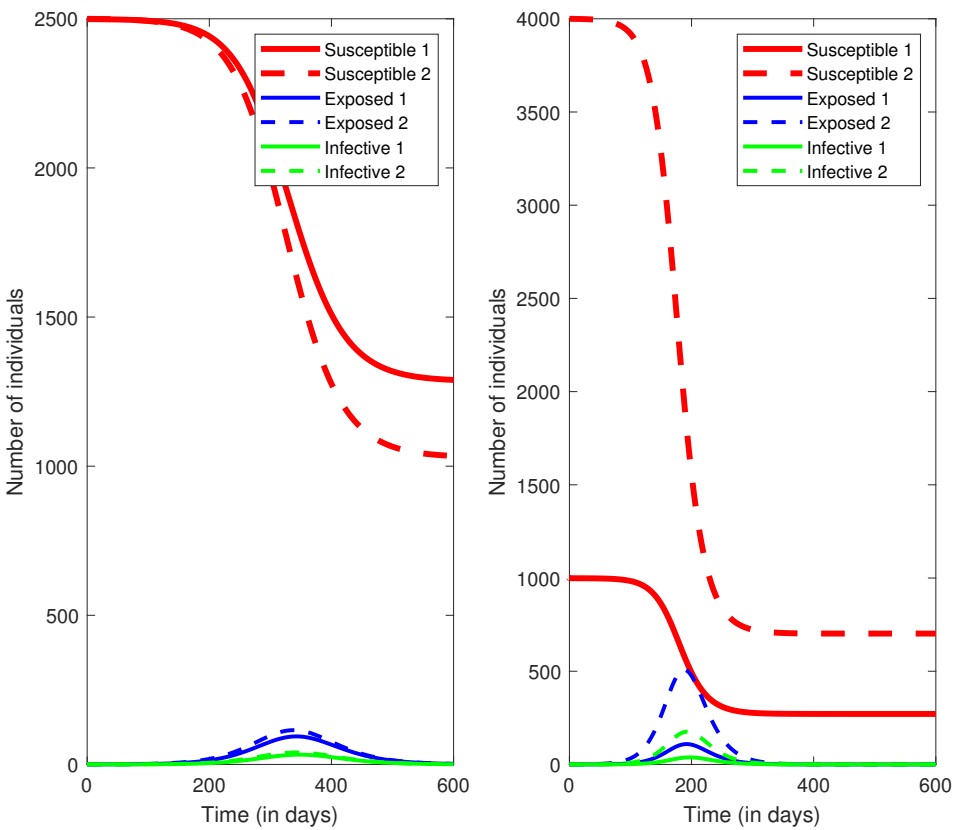

**Figure 1.** Simulations of the SEIR (Susceptible-Exposed-Infectious-Recovered) model (3) with two groups. The left sub-figure represents the proportion of face mask using $\rho_2 = 0.5$, and the right sub-figure represents $\rho_2 = 0.2$.

We then compared the attack rates for two groups when $\rho_2 = 0.2, 0.5$, respectively. The numerical results are summarized in Table 2. The attack rates for group 1 are always lower than group 2. If the proportion of face mask use increases, both groups benefit from lower probabilities of being infected.

**Table 2.** Attack rates for two groups with different proportions of face mask use.

| Attack Rates for Two Groups | | |
|---|---|---|
| **Face Mask Usages** | **Group 1** | **Group 2** |
| $\rho_2 = 0.2$ | 72.8% | 82.6% |
| $\rho_2 = 0.5$ | 48.4% | 58.6% |

### 3. Testing and Isolation of Diagnosed Infectives

In order to include testing and isolation of diagnosed infectives, we replaced model (3) with a model

$$S_1' = -\sigma_3 a S_1 \left[ \sigma_2 p_1 \frac{I_1}{N_1} + p_2 \frac{I_2}{N_2} \right],$$

$$S_2' = -a S_2 \left[ \sigma_2 p_1 \frac{I_1}{N_1} + p_2 \frac{I_2}{N_2} \right],$$

$$E_1' = \sigma_3 \beta S_1 \left[ \sigma_2 p_1 \frac{I_1}{N_1} + p_2 \frac{I_2}{N_2} \right] - \eta E_1,$$

$$E_2' = \beta S_2 \left[ \sigma_2 p_1 \frac{I_1}{N_1} + p_2 \frac{I_2}{N_2} \right] - \eta E_2, \tag{11}$$

$$I_1' = \eta E_1 - \gamma I_1 - p I_1,$$

$$I_2' = \eta E_2 - \gamma I_2 - p I_2,$$

$$Q' = p(I_1 + I_2) - \gamma Q$$

$$R' = \gamma(I_1 + I_2) + \gamma Q,$$

describing an *SEIR* model that includes testing of a fraction $p$ of the population, leading to the identification of $p(I_1 + I_2)$ infectives who are then isolated in a compartment $Q$. It is assumed that the total population size is a constant $N$, and that testing of $pN$ individuals yields $p(I_1 + I_2)$ positive tests. In fact, the number of infectives identified by testing of $pN$ individuals may be quite different from this; typically, tests are administered primarily to individuals thought to be infective. Moreover, there may be a significant delay between testing and identification. We assume that $p(I_1 + I_2)$ is the number of individuals identified as infective and quarantined. The control reproduction number for the model (11) is

$$\mathcal{R}_c = \Gamma_2 \frac{\beta}{\gamma + p}. \tag{12}$$

We can use our model to predict the number of disease cases for any choice of face mask usage and testing rates. We assumed a total population size of 1000. Current coronavirus testing in the US is at a rate of 600,000 tests per day, which corresponds to $p = 0.002$, but it has been argued by epidemiologists that the rate needs to be increased to at least 3,000,000 tests per day. For this reason, we used the values 0, 0.002, 0.01, and 0.02 for $p$ in our calculations, and we allowed $\rho_2$ to vary between 0 and 1. We continued to use the parameter values presented in Table 1.

The control reproduction number $\mathcal{R}_c$ for different values of $p$ and $\rho_2$ are summarized in Table 3.

**Table 3.** Dependence of control reproduction number $\mathcal{R}_c$ on face mask usage percentage $\rho_2$ and percentage of testing $p$.

| Summary of Reproduction Number $\mathcal{R}_c$ with Different $p$ and $\rho_2$ | | | | |
|---|---|---|---|---|
| **Face Mask Usages** | $p = 0$ | $p = 0.002$ | $p = 0.01$ | $p = 0.02$ |
| $\rho_2 = 0.5$ | 1.48 | 1.47 | 1.41 | 1.35 |
| $\rho_2 = 0.6$ | 1.28 | 1.27 | 1.22 | 1.16 |
| $\rho_2 = 0.7$ | 1.08 | 1.07 | 1.03 | 0.98 |
| $\rho_2 = 0.8$ | 0.88 | 0.87 | 0.83 | 0.80 |

There is a final size relation for model (11) from which we can calculate the size of the epidemic for each choice of the control parameters $\rho_2$ and $p$. The underlying point is that if the fraction of people using face masks consistently is increased to 80%, then the epidemic can be controlled.

## 4. Consequences

In order to control an epidemic in the absence of an available pharmaceutical intervention, it is necessary to decrease the reproduction number below 1. The numerical data obtained above indicate that in order to achieve this, it is necessary to decrease the contact rate to 40% of the "normal" rate, that is, to make the parameter $\Gamma_2$ no greater than 0.4. In order to achieve this using only testing and quarantine of infectives, it would be necessary to take $p = 0.3$, far beyond current possible testing rates. We suggest that increased use of face masks is the form of social distancing that is least disruptive to a population, and this is possible if $\rho_2 \geq 0.7$. Until now, government action to increase mask usage has been too weak to achieve this, mainly because the use of face masks has been transformed into a political issue, but increasing face mask usage to a value greater than 75% of the population should be achievable. The COVID-19 epidemic can be better controlled by current development and production of vaccines, by means of decreasing the fraction of susceptible individuals and the corresponding susceptibility.

## 5. A New Mutation

Very recently, a new mutated form of the novel coronavirus has appeared in several countries and is spreading widely [12]. It appears that the main difference of this mutation from previous forms is that it is considerably more infectious. It is believed that the infectivity is about 70% higher than previous forms, and this would suggest using parameter values $\beta = 0.85$, $\gamma = 0.2$ and $\eta = 0.07$. This indicates a basic reproduction number $\mathcal{R}_0 = 4.25$. Similarly, we performed two numerical simulations for cases of $\rho_2 = 0.2$ and $\rho_2 = 0.5$. The results are presented in Figure 2. It is observed that, with the larger value of $\mathcal{R}_0 = 4.25$, the times of infection peak are earlier and the total infection numbers are also increased. We summarize the results of attack rates for two groups when $\rho_2 = 0.2, 0.5$ in Table 4. The attack rates for both groups are very high. It is concluded that the more infectious mutated form of the novel coronavirus will cause a worse outcome if there is no extra public health measure being practiced.

**Table 4.** Attack rates for two groups with different proportions of face mask use when $\mathcal{R}_0 = 4.25$.

| Attack Rates for Two Groups | | |
|---|---|---|
| **Face Mask Usages** | **Group 1** | **Group 2** |
| $\rho_2 = 0.2$ | 92.7% | 96.7% |
| $\rho_2 = 0.5$ | 83.1% | 90.6% |

If we apply the reasoning of Section 3 to model (11) using the same parameters except for the new increased value of $\beta$, we find that in order to decrease the reproduction number below 1, we need to have $\Gamma_2 \leq 1/4.25 \approx 0.23$, and this can be achieved if the fraction $\rho_2$ of face mask users is more than 94.7%. Thus control of the more virulent form of the epidemic could be achieved if it is possible to persuade approximately 95% of the population to use face masks consistently. While the proportion of 95% might be unrealistic to achieve in practice, it is necessary implement social distancing measures to better control the spread of the epidemic. This also extends to the current outbreak models caused by holiday celebrations.

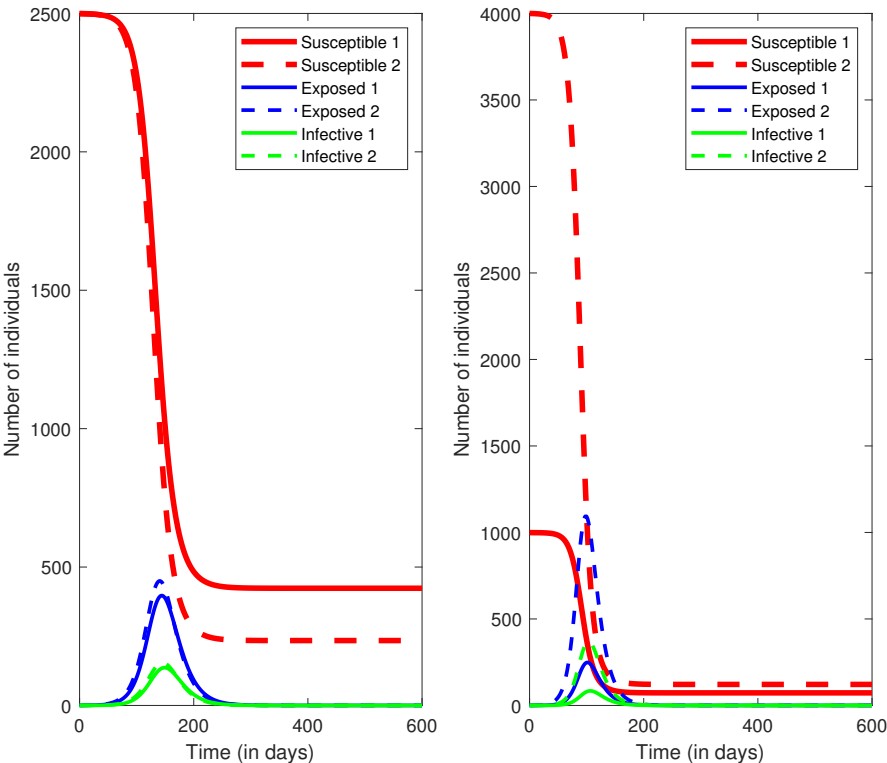

**Figure 2.** Simulations of SEIR (Susceptible-Exposed-Infectious-Recovered) model (3) with the larger value of reproduction number $\mathcal{R}_0 = 4.25$. The left sub-figure represents the proportion of face mask using $\rho_2 = 0.5$ and the right sub-figure represents $\rho_2 = 0.2$.

**Author Contributions:** Both authors contributed equally to this work. Both authors have read and agreed to the published version of the manuscript.

**Funding:** This research was supported by NSERC grant no. RGPIN-2016-03706.

**Institutional Review Board Statement:** Not applicable.

**Informed Consent Statement:** Not applicable.

**Acknowledgments:** F.B. (Fan Bai) acknowledges the postdocotoral fellowship supported from Hausdorff Center for Mathematics in University of Bonn.

**Conflicts of Interest:** The authors declare no conflict of interest.

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
