# Peer review of "The Effect of Face Mask Use on COVID-19 Models"

_epidemiologia, doi:10.3390/epidemiologia2010007_

Round 1

Reviewer 1 Report

Minor comments 

  • The authors should improve the readability of Figure 1, 2.
  • Some curves are not distinguishable so, they can be presented in the table as well.
  • It would be helpful to compare analytical and numeric results of the final epidemic size relation since the authors already obtained the final epidemic size relation (maybe the same for the controlled R0). 
  • It would be helpful to provide relevant references for the mutation section.

Author Response

Thank you for your suggestions. We have carefully read the comments and made changes correspondingly.

Reviewer 2 Report

Results are interesting, and sufficiently supported by the calculations. Unfortunately, there are some drawbacks in the presentation. Here are some of them:

  • The title is not appropriate: the paper is mainly about the use of masks, not generically about "Some Estimates of Epidemic Size for COVID-19 Models";
  • The discussion about the number of predicted cases in the US has nothing to do with the paper;
  • in particular, I don't understand why you first say that your prediction of almost 300M cases in US validates your model, and then immediately after you say that this is not an appropriate prediction;
  • similarly, the discussion about Economic effects of containment measures is not linked to the rest;
  • there is some confusion in the use of p_1, p_2, \rho_2 etc. From the context we have that p_1/N_1=p_2/N_2=1/N, p_1=\rho_2=N_1/N. So, why not use directly N_1/N with no extra variables?
  • Reference [2] for the final epidemic size is not adequate; perhaps [3] or [4] are the correct ones?
  • perhaps "attack rate" should be defined; is it the risk of infection?
  • there is some confusion on the value of N; sometimes it is 325M, in other places 5K.
  • I would also add a few references, for instance on schools:            Hyde, Z. (2020), COVID‐19, children and schools: overlooked and at risk. Med. J. Aust., 213: 444-446                                                 Gandolfi, A. (2021). Planning of school teaching during Covid-19. Physica D: Nonlinear Phenomena, 415
  • and on remote work, for instance                                                              Brynjolfsson, Erik, John J. Horton, Adam Ozimek, Daniel Rock, Garima Sharma, and Hong-Yi TuYe. COVID-19 and remote work: An early look at US data. No. w27344. National Bureau of Economic Research, 2020.
  • Also, Lines 33-34 seem to say exactly the opposite of Lines 35-36.

Author Response

(The authors gave the same response as above.)
